

# Areal rainfall estimation using moving cars – computer experiments including hydrological modeling

Ehsan Rabiei[1], Uwe Haberlandt[1], Monika Sester[2], Daniel Fitzner[2], Markus Wallner[3]

[1] [Institute of Water Resources Management, Hydrology and Agricultural Hydraulic Engineering, Leibniz Universität Hannover, Hanover, 30167, Germany]

[2] [Institute of Cartography and Geoinformatics, Leibniz Universität Hannover, Hanover, Germany]

[3] [Federal Institute for Geosciences and Natural Resources, Groundwater Resources - Quality and Dynamics, Hanover, 30655, Germany]

*Correspondence to*: E. Rabiei (rabiei@iww.uni-hannover.de)

**Abstract.** The need for high temporal and spatial resolution precipitation data for hydrological analyses has been discussed in several studies. Although rain gauges provide valuable information, a very dense rain gauge network is costly. As a result, several new ideas have been emerged to help estimating areal rainfall with higher temporal and spatial resolution. Rabiei et

al. (2013) observed that moving cars, called RainCars (RCs), can potentially be a new source of data for measuring rainfall amounts. The optical sensors used in that study are designed for operating the windscreen wipers and showed promising results for rainfall measurement purposes. Their measurement accuracy has been quantified in laboratory experiments. Considering explicitly those errors, the main objective of this study is to investigate the benefit of using RCs for estimating areal rainfall. For that, computer experiments are carried out, where radar rainfall is considered as the reference and the other

sources of data, i.e. RCs and rain gauges, are extracted from radar data. Comparing the quality of areal rainfall estimation by RCs with rain gauges and reference data helps to investigate the benefit of the RCs. The value of this additional source of data is not only assessed for areal rainfall estimation performance, but also for use in hydrological modeling. The results show that the RCs considering measurement errors derived from laboratory experiments provide useful additional information for areal rainfall estimation as well as for hydrological modeling. Even assuming higher uncertainties for RCs as

obtained from the laboratory up to a certain level is observed practical.

## 1 Introduction

Rainfall data is one of the most important information in hydrological analyses. The spatial and temporal resolutions of the data are crucial for the quality of hydrological analyses. Different modeling scales usually require different resolutions of





input data. A relatively high spatial and temporal resolution is required for smaller scale modeling such as in urban hydrology, whereas data with coarser resolution could be sufficient for larger scale hydrological modeling. Hypothetically, the performance of a model could be objectively judged when input data of a high quality is provided. In particular, the spatial and temporal resolution of rainfall amount over a study area influences the model performance significantly. The

quality of rainfall amount estimation depends, on the one hand, on the data availability, i.e. rain gauge network density, the temporal resolution of data and/or availability of additional information such as Digital Elevation Model (DEM) or radar data, and, on the other hand, on the interpolation techniques used for areal rainfall estimation.

Conventional rain gauges provide accurate point rainfall depth, but they are sparsely and irregularly located over the study area. This results in missing rainfall information where no rain gauge is available. On the other hand, a dense rain gauge

network is costly. There are several innovative ideas discussing new manners of measuring rainfall. Weather radar data with relatively high spatial and temporal resolution are widely used for rainfall estimation purposes, but the data are subject to several sources of error. Besides, weather radar is not available all over the world. Estimating rainfall using satellite data has become of interest for practical purposes, in particular for remote areas, because of good spatial coverage and being freely available. The satellite data suffers from the intrinsic weakness of the principle behind estimating rainfall, i.e. finding the

relationship between observable variables from space (e.g. cloud top temperature and the presence of frozen particles aloft) and rain intensity. Kidd and Levizzani (2011) and Kidd and Huffman (2011) have summarized some of the efforts given to improve the accuracy of satellite rainfall estimation. Several studies investigated rainfall estimation using microwave links as another potential source of data (Overeem et al., 2013; Rahimi et al., 2006; Upton et al., 2005; Zinevich et al., 2009), where a line-averaged precipitation is estimated therefrom. Acoustic rain gauges are an economical alternative which analyses the

raindrops sound similar to when one listens to rain in a tent (de Jong, 2010). Most of the mentioned studies seek for alternatives which either are not initially intended for rainfall estimation or have low operational costs.

Haberlandt and Sester (2010) hypothetically presented the potential of using moving cars for rainfall measurement purposes, called RainCars (RCs). They pointed at the potential of using RCs because of the widespread availability of cars especially in countries such as Germany. They concluded that a large number of hypothetically inaccurate devices could help in

improving the estimation of rainfall amount compared with just a few of accurate devices. Rabiei et al. (2013) investigated the possibility of using RCs for rainfall estimation with laboratory experiments. A strong relationship between rainfall intensity and the wiper speed, adjusted with front visibility, was observed. The rainfall estimation by the two optical sensors, Hydreon (2015) and Xanonex (2015), implemented in that study showed also promising results. Whether the derived accuracy of the sensors is sufficient for areal rainfall estimation or not is a question which is addressed in this study. Because

of the low number of real observations with RCs available on roads, the investigations are carried out by computer experiments. A continuous investigation using RCs with the derived uncertainties from laboratory experiments for a long period of time as well as implementing the data in a hydrological model would answer three important scientific questions: 1) Is the accuracy of optical sensors investigated by Rabiei et al. (2013) sufficient for areal rainfall estimation as well as





discharge simulation? 2) How relevant is the accuracy of rainfall data from RCs for areal rainfall estimation as well as for discharge simulation? and 3) What is the influence of using RCs over a longer period of time rather than just for certain events? These questions address the main objective of the study which is a better assessment of the value of the RCs for areal rainfall estimation rather than only for point measurement purposes.

The influence of input data quality on hydrological modeling performances has been under investigation by several studies. For example, Shrestha et al.(2006) investigated the influence of data resolution on the performance of a macro-scale distributed hydrological model (MaScOD). They split the factors influencing the quality of model performance into three categories: (1) the quality of the model, (2) the selected model parameters, and (3) the quality of the input data. The advantages of using RCs are assumed to provide a denser measurment network and additional information. Xu et al. (2013)

investigated the influence of rain gauge density and network distribution on the Xinanjiang River in China by the Xinanjiang Model. They found that the probability of getting a poor model performance increases significantly when the number of rain gauges falls below a certain threshold. They also concluded that the number of rain gauges above a certain threshold does not improve the model performance meaningfully. They realized that not only the number of stations is important, but also the spatial configuration of rain gauges.

This paper is organized as follows. The methodologies implemented in this study are presented after the "Introduction". Chapter 3 provides detailed information regarding the study area and data used in this study. The results and corresponding discussions are provided in chapter 4. Thereafter, a summary of the work and comparison of the results is presented with a more general conclusion.

## 2 Methods

Since there are not enough observed data from RCs, this study uses computer simulation. In order to carry out the analyses, rainfall fields as reference data are required. The point data from stations and RC's are extracted from the reference data and compared accordingly. There are essentially two possibilities to obtain reference data: (1) simulating the rainfall field or (2) using an available data source such as radar data. The latter choice has the advantage of being closer to reality and avoiding additional rainfall modeling. As a result, it is decided to consider radar data as reference and to extract the point data from

the radar data. As radar data has its deficiencies, the Mean Field Bias method is applied to correct the error in a straightforward way. The positions of RCs are provided by a traffic model and rainfall data are extracted from the reference data accordingly. The results are compared with what occurs in practice, i.e. using only the rain gauge network. The uncertainties for the rainfall measurement by RCs are taken from the results of the laboratory experiments (Rabiei et al., 2013). For a more general conclusion, larger uncertainties are also investigated.



## 2.1 Mean Field Bias correction

The Mean Field Bias (MFB) correction adjusts the radar data with the observed rain gauge data. Assuming that the rain gauge network provides accurate point precipitation data, the radar images could be corrected by:

$$B(t) = \frac{\sum_{i=1}^{n}\sum_{j=1}^{m}G(t_{i,j})}{\sum_{i=1}^{n}\sum_{j=1}^{m}R(t_{i,j})}$$

$$R^*(j) = B(t) \times R(j)$$

(1)

where $G(t_{i,j})$ is the precipitation amount from rain gauge $i$. $j$ is the time step within a time interval $t$. $R(t_{i,j})$ represents the precipitation amount on the radar pixel where rain gauge $i$ is located at the time $j$. $R$ and $R^*$ are the original and corrected radar rainfall, respectively. In this study, a daily time interval is considered for estimating the coefficient $B$ for each time step, which results in having a constant correction factor for each day, individually. This means that $m$ is 288 as the data are provided at a 5 minute temporal resolution. For the days on which Eq. 1 has an indeterminate form, i.e. when no rainfall

amount is recorded by the rain gauge network, the $B$ coefficient is set to 1.

Applying MFB does not have any smoothing effect, or, in other words, the structure of images after using MFB is very similar to that in the original radar data. Applying MFB to radar data was considered here to prevent unrealistic radar data values, whereas using radar data directly would also be possible since the relative errors obtained in the end would not change significantly.

## 2.2 Traffic model

The traffic model used in this study is similar to the one used by Haberlandt and Sester (2010). It is based on the road data derived from the Open Street Map (OSM). The traffic density is estimated using the data from the Federal Highway Research Institute (BASt) which provides the number of cars per day for certain points along *federal roads* and *highways*. For each particular catchment, those traffic count points within and close to the catchment, concerning *federal roads*

(corresponding to the OSM road type "primary"), are selected. The traffic count number per catchment is estimated therefrom. Based on this number, cars are generated applying the methodology described in the following. The assumptions underlying the traffic model are always conservative assumptions concerning the number and distribution of cars. This means that the number and spatial distribution of the cars is considered lower and less dense in the model than in reality:

    a)    Only larger roads on/and surrounding the study area are considered which includes the "*primary*" and "*secondary*"

OSM road types (corresponding to the German "Bundesstraßen" and "Landstraßen" road types). Because of the





relatively high practical uncertainties related to the RCs on *highways*, these road types are excluded. Smaller roads are also neglected due to the low traffic.

b) An average speed of 80 km/h is considered to calculate the number of cars for each catchment. The assumed average speed is higher than in practice, which results in a lower number of cars than in reality (see Eq. 2). This follows the conservative assumption mentioned earlier.

c) Due to the lack of traffic count data for the "*secondary*" OSM road type, the traffic count for this road type is calculated using half the "*federal roads*" (OSM "primary" roads) traffic count data. This also follows the conservative considerations for the traffic model assumptions.

In order to estimate the number of cars driving simultaneously within and around a catchment, the following equation is used for each catchment and each of the two road types separately:

$$
\begin{aligned}
t &= \frac{X}{h} \\
z &= t^{-1} \cdot \bar{v} \\
n &= \frac{l}{z}
\end{aligned}
\tag{2}
$$

where $X$ is the number of cars from the traffic data over a certain time period $h$, $\bar{v}$ is the assumed average car speed and $z$ is the space between two cars. The number of cars driving simultaneously in and around the catchment area $n$ is then estimated using the total roads length $l$ on a catchment. Due to the long period of time considered in this study, the day-night variation in traffic count is considered insignificant. Therefore, the daily average number is used in this study. This number is subsequently used for generating cars randomly on the OSM road network at each time step. This means that the points representing RCs are not dependent in successive time steps, i.e. no car identities are modeled.

**2.3 Network density of rain stations**

In order to compare the network densities for rain gauges and RC scenarios, the network density of each subcatchment for each scenario is calculated in a similar way to that used by Haberlandt and Sester (2010). The network density is calculated using the kernel density estimator (Silverman, 1986):

$$
D_i = \frac{1}{\pi r^2} \sum_{j=1}^{n} k_j \quad \text{with} \quad k_j = \begin{cases} 3\left(1 - \left(\dfrac{d}{r}\right)^2\right)^2 & \text{for } d \leq r \\ 0 & \text{for } d > r \end{cases}.
\tag{3}
$$





where $n$ is the number of observation points (either stations or RCs) within the search radius $r$ and $d$ is the distance to subcatchment cells (the ones for which the density is being calculated). $D_i$ is calculated for each subcatchment cell and averaged over all subcatchment cells. The kernel density estimator considers not only the observation points in the subcatchment, but also the ones within the search radius.

## 2.4 Uncertainties for RainCars

In order to consider the uncertainties in rainfall measurement using RCs, the results of laboratory experiments (Rabiei et al., 2013) are utilized. The relationship between sensor reading ($W$) and rainfall intensity ($R$) is named W-R relationship. Signal lengths from the optical sensors are considered as sensor readings.

$$\hat{R} = a + bW + \varepsilon \tag{4}$$

where $\widehat{R}$ is the rainfall intensity, $W$ is the sensor reading, $a$ and $b$ are the linear regression coefficients and $\varepsilon$ represents the random error. The assumption behind the linear regression model is that the error is normally distributed, with mean = 0 and variance = $\sigma^2$. This provides a simple error model for the measured uncertainties from RCs.

Three different linear W-R relationships between sensor readings and rainfall amount are discussed by Rabiei et al. (2013). The first relationship considers wiper frequency as sensor. As wiper activity is influenced by several factors, such as driver preferences, car speed, number of wiper speed levels defined for each car type, etc, this alternative is considered as impractical. The two other remaining alternatives are the W-R relationships derived from the optical sensors. Two optical sensors, Hydreon and Xanonex, with promising results were investigated by Rabiei et al. (2013) and suggested for further use. The two sensors performed similarly, whereas the Hydreon performed slightly better. Because of the Xanonex shape and its ease of installation on cars, it is decided to investigate the Xanonex W-R relationship.

Although a relatively strong relationship between the two variables exists, one encounters difficulties for smaller rainfall amounts. According to the estimated regression line, negative rainfall can be calculated, which is not possible. By neglecting the negative values, a systematic positive bias would enter the data. On the other hand, the uncertainties of the devices on the market are usually expressed as a percentage, which illustrates a smaller absolute error when smaller values are measured. Considering those originally obtained linear relationships does not account for this and calculating implausible rainfall amounts are possible. Therefore, in order to investigate the uncertainties for the sensor readings, a power regression model describes the W-R relationship:

$$\hat{R} = a \cdot W^b \cdot \varepsilon \tag{5}$$





where $\hat{R}$ is the rainfall intensity, $W$ is the sensor reading, $a$ and $b$ are the regression coefficients and $\varepsilon$ is the random error. Taking the logarithm of both sides of Eq. 5 gives:

$$\log(\hat{R}) = \log(a) + b \cdot \log(W) + \log(\varepsilon) \tag{6}$$

As is assumed for simple linear regression, a constant random error, here $\log(\varepsilon)$, is considered. It should be noticed that the error variance is constant in log transformed space and variable in original space.

Negative rainfall amounts can no longer be estimated because of the log-log transformation to the data. Implementing this data transformation also leads to a more accurate performance for smaller rainfall values, which is different to the constant value considered by Rabiei et al. (2013).

The traffic model provides coordinates of the RCs for each time step. The rainfall amount for each RC is extracted from the reference data set, i.e. radar data. The device outputs are signal lengths related to rain intensities (Rabiei et al., 2013). In order to consider the uncertainties for RCs, the corresponding signal length of each extracted value would be estimated using the W-R relationship. A normally distributed error $\log(\varepsilon)$ with mean = 0 and variance = $\sigma^2$ is randomly selected and added to $\log(R)$ before re-transformation.

**2.5 Areal rainfall estimation**

Ordinary Kriging (OK) is an interpolation method which is widely used for several hydrological variables such as temperature, rainfall, wind etc. OK is implemented here for interpolating data from both RCs and rain gauges. The benefit of using RCs can be explored by comparing the quality of areal rainfall estimated by rain gauges with when only RCs are used instead. For a detailed description of the method, please refer to geostatistical text books such as Isaaks and Srivastava (1990).

The experimental variogram is estimated using the following equation:

$$\gamma_k(h) = \left[ \frac{1}{2 \cdot N(h)} \sum_{i=1}^{N(h)} (Z(x_i) - Z(x_i + h))^2 \right] \tag{7}$$

where $N(h)$ represents the number of data pairs, $h$ is the separating vector, $x$ the location and. Similar as in Rabiei and Haberlandt (2015), a seasonal average variogram is used here. The experimental variogram is estimated using radar data when 1000 random radar cells are taken. Only the time steps with an average rainfall above a defined threshold are selected for variogram estimation. The following equation is used for estimating average variograms over n time steps:



$$\gamma_{st}(h) = \frac{1}{n} \cdot \sum_{i=1}^{n} \frac{\gamma(h,i)}{\text{var}(i)} \qquad (8)$$

where $\gamma(h,i)$ is the variogram for the $h$ distance class and var($i$) represents the variance in time step $i$. An exponential variogram is considered as the theoretical variogram model:

$$\gamma_h = c_0 + c\left[1 - \exp\left(-\frac{h}{a}\right)\right] \qquad (9)$$

where $a$, $c$ and $c_0$ are the range, the sill and the nugget effect, respectively.

The variograms are fitted using radar data with 5 min temporal resolution as the goal is to interpolate rain gauges as well as RCs on 5 min temporal resolution.

**2.6 The HBV hydrological model**

The hydrological model used in this study, HBV-IWW, is a modified semi-distributed version of the HBV model (Lindström
et al., 1997). The model has a horizontal spatial discretization in subcatchments, which are linked to each other by river reaches. For each of the subcatchments a snow routine, a soil routine, a response routine and a transformation routine is applied. The snow routine classifies precipitation as rainfall or snowfall and also takes snow melt into account. After that, the sum of the rainfall and snowmelt passes the soil routine which consists of two modules. The first module calculates the actual evapotranspiration, while the second module calculates the contributing runoff depending on precipitation and actual
soil water content. The contributing runoff is then directly linked to the upper groundwater layer of the response routine, where surface runoff, interflow, percolation and the actual water content of the upper groundwater layer are calculated. Percolation contributes to the lower groundwater layer wherefrom the base flow is calculated. Surface runoff, interflow and base flow are finally added together and transformed with a simple triangular unit hydrograph. If more subcatchments are connected to each other, the Muskingum method is used for river routing.
The model is calibrated using the Simulated Annealing algorithm (Kirkpatrick, 1984) for which 1000 iterations are considered. The objective function is:

$$OF = (1 - NSE) + (1 - NSE_{Log}) \rightarrow \min \qquad (10)$$

where $NSE$ is the Nash-Sutcliffe coefficient after Nash and Sutcliffe (1970), and $NSE_{Log}$ is the $NSE$ with logarithm of discharges. A more detailed description of the parameter calibration procedure as well as further details of the HBV-IWW
model can be found in Wallner and Haberlandt (2015). Unlike the common procedure of calibrating the parameters of a hydrological model and validating them afterwards, when two separate time periods are defined, in this study the whole time



period is considered for calibrating the model parameters. The HBV parameters are calibrated lumped as only the rainfall data are to be investigated. This means that all the subcatchments of each catchment have the same model parameter set. For all the scenarios in the following, the same parameter sets are used for an explicit comparison of the results. As the main purpose of the study is to investigate the influence of different means of rainfall measurement, the model calibration is less
important than in studies dealing with observation data.

## 2.7 Performance measures

A common way to evaluate the performance of interpolation is  cross-validation, i.e. the leave-one-out approach. The resemblance of the estimations to the observations illustrates the quality of the interpolation technique. Since reference radar data are considered as the truth in this study, the areal rainfall estimated by each scenario is directly compared with the
reference areal rainfall. The following criteria are used for evaluation.

The Root Mean Square Error is estimated by:

$$RMSE(i) = \frac{1}{J} \sum_{j=1}^{J} \left[ \sqrt{\frac{(Z^{*}_{i,j} - Z_{i,j})^2}{n}} \right],$$    (11)

the Nash-Sutcliffe coefficient by:

$$NSE(i) = 1 - \frac{\sum_{j=1}^{J} (Z^{*}_{i,j} - Z_{i,j})^2}{\sum_{j=1}^{J} (Z_{i,j} - \overline{Z})^2}$$    (12)

and the Percent Bias (*Pbias*) is estimated by:

$$Pbias(i) = 100 \times \frac{\sum_{j=1}^{J} (Z^{*}_{i,j} - Z_{i,j})}{\sum_{j=1}^{J} (Z_{i,j})}$$    (13)

where $Z^{*}$ is the estimated areal rainfall and $Z$ is the corresponding reference areal rainfall. $j$ is the number of time steps considered for the subcatchment $i$. These statistical measures are used also for evaluating the performance of the hydrological model where $Z^{*}$ and $Z$ are then the simulated discharges and reference discharges, respectively.
A positive *Pbias* indicates overestimation, whereas a negative value indicates underestimation.



## 3 Study area and data

A part of the state of Lower Saxony covered by the weather radar located at Hanover airport and the three catchments in Fig. 1 encompass the study area. As mentioned, the benefit of RCs is investigated by comparing with what occurs in practice, i.e. when only rain gauges are considered. In this study, it is assumed that the coordinates of rain gauges and the coordinates of

53 rain stations provided by the German Weather Service (DWD) are identical.

- **Fig. 1.**

The transparent blue circle in Fig. 1 with a 128 km radius is the area being scanned by the Hanover weather radar, whereas the points represent the 53 rain gauges considered in this study. The Digital Elevation Model shows that the northern part of the study area is relatively flat and a region with mountainous characteristics is in the south-eastern part. The precipitation

amount also varies within the study area from around 500 mm/yr in the north to 1700 mm/yr in the mountains. The mean annual rainfall in each subcatchment shows also the spatial rainfall variation over the study area. This is more evident for the Nette catchment as the south-eastern subcatchment receives a larger amount of rainfall than the other subcatchments. Although the Sieber catchment is located in the mountainous area, the mean annual rainfall is relatively low. This could be explained by the fact that the catchment is located at the leeward side of the mountains considering the usual west-to-east

weather front moving direction.

### 3.1 Catchments

Three catchments of the Aller-Leine river basin, which have different characteristics, are considered in this study, Fig. 1. Not only are the characteristics of the catchments important, but also are the locations of the rain gauges. The Böhme catchment located in the northern part, with a relatively flat terrain, contains eight subcatchments and covers 285 km$^2$. This catchment

varies between 50 m and 150 m in elevation and contains one rainfall station. The Nette catchment has 10 subcatchments covering 309 km$^2$ and is partly located in the mountainous area, where the elevation reaches up to 550 m. In contrast, the northern part of this catchment is mostly flat. An important point worth mentioning here is that the only station available in this catchment is located in front of the hillside in the southern part. The Sieber catchment is located completely in the mountainous area and has two subcatchments covering 45 km$^2$. There are some stations close to the catchment, but no

stations are available within it.

### 3.2 Radar rainfall

The C-band Hannover weather radar provides radar data with a 5-min temporal resolution and $1° \times 1$ km spatial resolution. The time period from 2006 to 2010 is considered in this study. The dx-radar product provided by the DWD is used and





processed as following. First, the reflectivity ($Z$ in mm$^6$m$^{-3}$) is transformed to rain intensity ($R$ in mm/hr) by the following relationship:

$$Z = a \cdot R^b .\tag{14}$$

Standard DWD parameters (Riedl, 1986; Seltmann, 1997) are used, where $a = 256$ and $b = 1.42$. A straightforward clutter detection similar to that of Berndt et al. (2014) is applied thereafter. The final step is to interpolate the rain intensities on rectangular grids using the Inverse Distance Weighted (IDW) technique. This produces rainfall of 1 km × 1 km spatial resolution. Afterwards, the Mean Field Bias method (see section 2.1) is implemented to adjust radar data with the observed rain gauge data.

As the observed data are not used directly for the objectives of this study, it is decided not to describe them here to avoid any confusion.

### 3.3 W-R relationship

Rabiei et al. (2013) used a linear regression model to describe the W-R relationship between the Xanonex sensor readings and rain intensity. Fig. 2 illustrates this relationship. The dots represent the observations in the laboratory whereas the dashed lines show the 95% prediction limits. A detailed description of the laboratory experiments is provided in Rabiei et al. (2013). The main disadvantage is when facing small rainfall values. As mentioned, by considering this relationship and the error distribution for linear regression, negative rainfall amounts can be estimated.

- **Fig. 2.**

### 4 Results and discussion

In the following, the results of the steps taken for investigating the benefit of using RCs for areal rainfall estimation as well as discharge simulation are presented and discussed.

### 4.1 Traffic model

The number of cars is estimated using Eq. 2. It is assumed that only a small portion of cars is equipped with sensors measuring rainfall. In this study, from 1% to 5% of all cars on the roads are considered to measure rainfall which describes all the RC scenarios. For each 5-minute time step, the number of cars is calculated for the 5% scenario. The other scenarios are generated therefrom. Table 1 depicts different RCs' scenarios considered in this study.

- **Table 1.**





Fig. 3 shows the road network considered for the three catchments. As seen in Table 1, a denser network than for the other catchments is available for the Nette catchment. As the Nette catchment is partly located in mountainous area, even this denser network might not provide enough information. This is due to the fact that the RCs are not available overall because of the road network. For that reason, in subcatchments such as the south-eastern subcatchment, the number of available RCs

is lower than in the other subcatchments.

- **Fig. 3.**

## 4.2 Network density

Before illustrating the results of the simulations, i.e. areal rainfall as well as runoff simulation comparison, the network densities using Eq. 3 for different scenarios are presented. This helps to determine whether the network density influences

the results.

Fig. 4 shows the network densities estimated for the scenarios being investigated in this study. Although the density varies among the catchments, Fig. 4 shows that all the RC scenarios have a higher density than the rain gauge network. Depending on the accuracy of the measurement devices, i.e. RCs or rain gauges, the network density has a variable influence. A more detailed investigation is provided in subsection 4.6.2.

- **Fig. 4.**

## 4.3 RainCars Uncertainty

The uncertainty related to rainfall estimation by RCs is described by Eq. 4 to Eq. 6 where $\varepsilon$ represents the random error. The normal distribution that defines the random error for each signal reading (signal length) corresponding to each rainfall amount has the residual variance estimated by the vertical distances between the observations and the regression line. The

random error, $\log(\varepsilon)$, is then simulated using the normal distribution. As discussed earlier, a power regression model describes the W-R relationship in this study.

Fig. 5 (a) shows the W-R relationship after log-log transformation. The same assumption as before is valid, namely that the random error is normally distributed and derived from the deviation between observation points and the linear regression model. Fig. 5 (b) illustrates the W-R relationship implemented in this study. It is derived using the following steps: 1)

applying log-log transformation on both axes, 2) applying linear regression on the transformed data (Fig. 5 (a)), 3) estimating the residual variance for the normal distribution describing the random error for the linear regression model and 4) transferring the data back for practical use (Fig. 5 (b)).

- **Fig. 5.**





The data transformation has, in general, two important effects on the W-R relationship: 1) preventing the estimation of unrealistic rainfall amounts and 2) skewing the distribution of random error. The latter aspect affects also the prediction limit in Fig 5. As can be seen in Fig. 5b, the upper and lower limits bend when further from the origin which results in larger inaccuracy for the rainfall amount estimated by RCs for higher rainfall intensities. On the other hand, the positive skewness

introduces a positive bias that causes overestimation when estimating rainfall amount by RCs. This can be seen when comparing the distances from the model line to the upper and lower prediction limits. Although the W-R relationship has this deficiency, a larger number of RCs and more accurate optical sensors can help compensate this problem. These two aspects are addressed in section 4.6.2. and 4.6.3, respectively.

## 4.4 Variogram properties used in this study

The properties of the variograms used in this study are provided in Table 2.

- **Table 2.**

The variograms are fitted using radar data with 5 min temporal resolution as the goal is to interpolate rain gauges as well as rainfall from RCs on a 5 min temporal resolution. A relative large range is estimated in winter time which illustrates different seasonal rainfall patterns. It supports the seasonal separation for interpolating the data which was discussed earlier.

As can be seen, the properties of the variograms change even among the same seasons in different years. Therefore, it is decided to use the variable variograms provided in Table 2.

## 4.5 Reference discharge

As mentioned earlier, the simulated discharge for different scenarios will be compared with the reference discharge. The reference discharge, the benchmark, is simulated using radar data after applying Mean Field Bias correction (creating the

reference rainfall data) as input to the HBV-IWW model using pre-calibrated model parameters. Because of the lumped approach for calibrating the model parameters and the subcatchments with relatively small size, the rainfall characteristics, especially its spatial pattern, are the highest influencing factor in discharge simulation.

The performance of the HBV-IWW model is evaluated on an hourly temporal resolution. Therefore, an aggregation of 5 min interpolation data to hourly data is carried out before using it in the hydrological model.





## 4.6 Comparing areal rainfall and simulated discharge for different sources

The value of the RCs in comparison to the rain gauge network is assessed by comparing areal rainfall estimations as well as the simulated discharges using these data. First, the results of using only the rain gauge network are presented. Thereafter, the results of using RCs for rainfall observations are provided and compared with when only the rain gauge network is used.

### 5 4.6.1 Rain gauge network

**Areal rainfall estimation**

The areal rainfall estimations corresponding to the three catchments shown in Fig. 1 are compared with the reference data. It should be noticed again that the comparison is carried out after interpolating the data with 5 min temporal resolution and aggregating the data to hourly temporal resolution because of the required temporal resolution for the hydrological model.

10 Fig. 6 provides the statistical measures for the three catchments under study when evaluating the quality of areal rainfall estimation using only rain gauges. This is carried out by comparing the estimated areal rainfall using rain gauges and implementing OK with the reference data.

- **Fig. 6.**

For the Böhme catchment having one station in the catchment and one close by provides sufficient rainfall information. As 15 expected, the closer the subcatchments to the stations, the better the quality of areal rainfall estimation. The areal rainfall estimated for the northernmost subcatchment is not as good as for the other subcatchments because there is no station nearby. Although the potential for improving *RMSE* and *NSE* values exists, the *Pbias* criterion is in general relatively low. This means that the total water volume is estimated relatively well, and therefore for purposes such as hydrological modelling the quality of areal rainfall estimation might be sufficient.

20 Studying the Nette catchment, the subcatchment that includes a station has, as expected, a superior rainfall estimation quality to the other subcatchments. Unpredictably, the quality of areal rainfall estimation for the other subcatchments close to the station is weak. For example, although the two southern subcatchments are in the vicinity of a station, the statistical measures are relatively poor. A rapid change in elevation is evident when considering the DEM map in Fig. 1. Assuming that rainfall characteristics change along the elevation gradient, a change in the spatial rainfall pattern is expected. The single 25 station is no longer able to provide the actual rainfall amount even for the surrounding subcatchments. This is in contrast to the Böhme catchment where the DEM map shows a flat catchment and the only station on the Böhme catchment is sufficient for areal rainfall estimation.



Although the Sieber catchment is smaller than the other two catchments and is expected to be more easily modelled, the catchment is located in a mountainous region and suffers from the fact that no rain gauge is available directly within the catchment. Owing these facts, the areal rainfall estimation is rather poor, especially when the *Pbias* is of concern. For such conditions, additional means of rainfall measurement would be beneficial.

**5   Discharge simulation**

The same statistical measures are used for evaluating the performance of the hydrological model. Depending on the location of stations, catchment characteristics and spatial rainfall pattern, each catchment responds differently when only rain gauges are used. The reference discharge, the benchmark, is simulated using reference areal rainfall, i.e. radar data after MFB, and the pre-calibrated model parameters.

Table 3 provides the statistical measures of simulated discharges when only rain gauges are implemented. Although both the Böhme and Nette catchments benefit from having a station in the catchment, the two catchments responded differently. The Böhme catchment performs better than the other two catchments. From Fig 7, it can be seen that the quality of the areal rainfall estimation for the Böhme catchment is the best. As seen in the *study area and data* section, the mountainous area receives more rainfall than the other parts of the catchment. The mountainous part of the catchment can cause a change in the spatial rainfall pattern. In other words, a fast elevation change (when the contour lines are tightly spaced together) can draw the isohyetal lines close together. This can explain the reason that the model performance in the Nette catchment is relatively poor. From the mean annual rainfall amount for each subcatchment provided in Fig. 1, it can also be concluded that the two southern subcatchments in the mountainous area produce a big share of the discharge. It is observed that one station can be sufficient for areal rainfall estimation for a flat catchment such as the Böhme catchment and would not be sufficient for a catchment such as the Nette catchment, which is partly located in the mountainous area. There is no station located in the Sieber catchment, which explains the poor *Pbias* in Table 3, although the other two criteria are rather good. In the Sieber catchment, not only the characteristics of the two subcatchments are similar, but also the spatial rainfall pattern over the two subcatchments which could explain the relatively good *RMSE* and *NSE* criteria.

- **Table 3.**

**4.6.2 RCs against rain gauges using errors from laboratory experiments**

**Areal rainfall estimation**

A similar strategy is pursued for the moving cars measuring rainfall. As before, the evaluation involves two parts. First, the areal rainfall estimation by implementing RCs is compared with the reference rainfall. Thereafter, the simulated discharges are compared after using the data in the HBV-IWW hydrological model.



The benefit of using RCs for areal rainfall estimation can be assessed when their performance is compared with the standard approach, i.e. using only the rain gauges. To this end, after estimating the statistical measures by comparing with reference data, the difference between the statistical measures is addressed. This means that for example for the Root Mean Square Error $RMSE_{diff} = RMSE_{RCs} - RMSE_{St}$. As a result, negative $RMSE_{diff}$ values as well as positive $NSE_{diff}$ values represent better

areal rainfall estimation when using RCs compared to stations. For *Pbias*, the specific values are compared without building differences.

Fig. 7 illustrates the statistical measures for the Böhme catchment when the RCs are used for areal rainfall estimation. Implementing RCs in general results in better areal rainfall estimation. As expected, the improvement in subcatchments away from the stations is more significant than the ones close to them. Also, the number of cars plays an important role. If

the number of RCs increases, the quality of areal rainfall estimation improves. Considering only the two mentioned criteria, using RCs for areal rainfall estimation is always superior to using stations in this catchment. The improvement for each subcatchment varies depending on the number of RCs as well as the location of the station. $Pbias_{RCs}$ values show an overestimation of areal rainfall because of the positive skew of the error distribution as explained earlier (see Fig. 5b).

- **Fig. 7.**

Fig. 8 illustrates the statistical measures for different RC scenarios in the Nette and Sieber catchments.

The use of RCs for rainfall estimation in the Nette catchment has similar advantages as for the Böhme catchment. In contrast to that, RCs are not always beneficiary in the Nette catchment. A detailed investigation shows that the rainfall estimation for the subcatchment in which the station is located is hard to beat by using RCs. In contrast to all the subcatchments where an overestimation is observed, for the Nette basin, the subcatchment with red colour in *Pbias* responds differently. This can be

explained by the RC network density. This part of the catchment suffers from the fact that RCs are rarely available because there are fewer roads as it is located in the mountainous part.

For the Sieber catchment using RCs also results in better areal rainfall estimation. Increasing the number of RCs has again advantages. The improvement in areal rainfall estimation is not as strong as in the other catchments because of the existing stations in the vicinity of the catchment. Although *RMSE* and *NSE* do not vary significantly, comparing with Fig. 6, the

*Pbias* criterion improves meaningfully.

It should be noted that although the Nette catchment benefits from a denser RC network (Fig. 4), because of the spatial rainfall pattern, the need for a higher number of RCs or a better location for the only station is evident.



- **Fig. 8.**

**Discharge simulation**

Table 4 provides the statistical measures of simulated discharges when RCs are implemented. The first column, titled "St.", refers to when only rain gauges are considered, which is included here again to facilitate easy comparison.

5 - **Table 4.**

Although the Böhme catchment performs the best among the three catchments when only rain gauges are considered, using RCs is still useful. Fig. 8 shows that the areal rainfall estimation improves slightly when using RCs. For analyses requiring fine temporal and spatial resolution data, e.g. urban hydrology, using RCs may improve the simulation results more evident. Due to the fact that the improvement in discharge simulation is not very strong in this catchment, with the given temporal 10 and spatial resolution, for such studies the need for using RCs can be considered inessential.

As discussed earlier, because of the characteristics of the Nette catchment, this catchment has the highest potential for improvement by RCs. Implementing even a small number of RCs improves the results significantly. As seen in Fig. 4, the Nette catchment has the highest RC network density among the three catchments. This explains the better performance in this catchment. As expected, positive Pbias in the simulated discharge indicates overestimation, which follows the areal 15 rainfall overestimation observed earlier.

Unlike the other catchments, the Sieber catchment is located in the mountainous area. The NSE and RMSE criteria do not improve significantly when using RCs. Taking a deeper look at the traffic model data, from 1% scenario to 5% scenario, the number of cars measuring rainfall is 3, 6, 8, 11 and 14, respectively (see Table 1). Only two scenarios can result in better discharge simulation, 4% and 5% indicating the least required RC network density for this catchment. As with the other two 20 catchments, using RCs results in the overestimation of the discharge.

The network density of the 1% RCs scenario for the Nette catchment is similar to those of the Böhme and Sieber with 2% RCs scenarios (Fig. 4). Taking the similarity of the network densities into account, the improvement of the hydrological model performance in the Nette catchment is more evident than for the other two catchments. It shows that the RCs are more valuable in the catchments such as the Nette when the spatial rainfall pattern varies within the study area (see section 3). 25 Improving the discharge simulation performance for bigger catchments such as the Nette and Böhme seems to be easier achievable with less density of RCs than for smaller catchments such as the Sieber. Additionally, even a small number of RCs can improve the discharge simulation significantly. Depending on the quality of the required hydrological analyses, the



need for the use of RCs is open to discussion. Basically, a higher number of equipped cars are needed for mountainous and/or small-scale study areas than for big and flat study areas.

### 4.6.3 RCs against rain gauges using hypothetical errors

**Areal rainfall estimation**

The minimum rainfall measurement accuracy required for the RCs to be useful is investigated in this section. All the different accuracies are addressed for the 5% RCs scenario specifically.

The error for the linear model is estimated on the log-log transformed data, Fig. 5. The normal distribution was defined by the variance ($\sigma^2$), equal to 0.021, from laboratory experiment results. In order to investigate the importance of the error on the results, four other variances of 0.0, 0.01, 0.04 and 0.09 are considered. At the end, the areal rainfall estimation quality as

well as the performance of the hydrological model was compared with that of using the original variance from the laboratory.

Table 5 provides the averaged statistical measures for each catchment. The areal rainfall estimation considering different errors for RCs is compared with the reference data. The same assumptions as earlier (see section 4.3.) are taken with different variances for the distribution function representing the error range. St. represents the areal rainfall estimation

performance when only the rain gauges are implemented.

- **Table 5.**

The rainfall overestimation by implementing higher $\sigma^2$ values is evident. For all catchments even assuming a relatively large uncertainty of $\sigma^2 = 0.09$, *NSE* and *RMSE* values improve compared with when only rain gauges are considered. As the *Pbias* is quite large for $\sigma^2 = 0.04$ and $\sigma^2 = 0.09$, the use of RCs for areal rainfall estimation is questionable. In fact, for such cases,

using RCs as additional information for areal rainfall estimation, e.g. when using External Drift Kriging or Kriging with Uncertain Data, can be more useful.

Assuming no inaccuracy for the measurement devices, i.e. St. and $\sigma = 0.0$, a negative *Pbias* still exists representing an underestimation of areal rainfall. Although OK is an unbiased interpolation technique, its performance is strongly dependent on the measurement locations. In an ideal situation, measurements should take place in regard to the variation in spatial

rainfall patterns. This can not be fulfilled in practice due to the dynamic nature of rainfall. Missing the minima and maxima over the study area can lead to overestimation and underestimation, respectively. It is more probable to miss maxima than minima due to the fact that high rainfall intensities may occur in places where no RCs or rain gauge observations are





available. Minima can be captured easier than maxima as it covers a larger area. This might explain the minor negative *Pbias* derived by RCs even if when σ = 0.0.

**Discharge simulation**

Table 6 provides the averaged statistical measures of the simulated discharges for each catchment when the 5% RC scenario is considered. As expected, the best performance belongs to the RCs scenario for which the measurement error is assumed to be zero ($\sigma^2 = 0.0$). The quality of the simulated discharge lowers by increasing the error of the measurement devices (RCs). A similar trend can be found as that for areal rainfall estimation, in that the discharge overestimation becomes meaningful by increasing the uncertainty of the RCs. Implementing RCs with large uncertainty for the measurement values leads to a relatively weak discharge simulation. On the other hand, even though using RCs results in discharge overestimation (*Pbias* criterion), the quality of simulated discharges for variances ($\sigma^2$) smaller than 0.04 improves in terms of *RMSE* and *NSE*, compared with when only rain gauges are considered (St.). As discussed before, in order to overcome the overestimation caused by RCs, one may consider RCs as additional information in interpolation techniques. RCs could be corrected in practice by implementing quantile mapping like the one introduced by Rabiei and Haberlandt (2015).

- **Table 6.**

As observed, the improvement of model performance in the Nette catchment when using RCs is more evident than for the other two catchments. Therefore, it is decided to investigate the Nette catchment in more detail by analysing the hydrographs for all the scenarios given in Table 6.

Fig. 9 shows the simulated discharge for all the scenarios provided in Table 6. It can be observed that when only rain gauges (St.) are used, the model misses some peaks and performs poor. This illustrates that the local rainfall is often not captured. The model performance improves significantly when using RCs. By increasing the uncertainties, i.e. enlarging $\sigma^2$, the overestimation of rainfall amount affects the model performance as well.

- **Fig. 9.**

**5 Summary and conclusion**

The value of using moving cars for rainfall measurement purposes (RCs) was investigated with laboratory experiments by Rabiei et al. (2013). They analysed the Hydreon and Xanonex optical sensors against different rainfall intensities. The optical sensors showed promising results when used for point rainfall measurement. Because of the low number of real RCs





available on roads, the main objective of this study was to implement and investigate the errors derived from the laboratory experiments for areal rainfall estimation in a computer simulation. The errors were considered for the theoretical RCs, provided by a traffic model, and Ordinary Kriging (OK) is implemented for areal rainfall estimation. Thereafter, the data are also used for discharge simulations in the HBV hydrological model. The value of the RCs is compared with when only rain gauges are implemented. Radar data was considered as the reference data to directly evaluate the areal rainfall estimation rather than following the common approach for evaluating an interpolation technique, i.e. cross-validation. The other sources of data, i.e. RCs and rain gauges, were extracted from the reference data source, accordingly. A period of 5 years from 2006 to 2010 and three catchments with different characteristics are considered.

The results of the study are as follows:

1) Implementing RCs with the uncertainties derived from the laboratory experiments improves the quality of areal rainfall estimation compared with when only rain gauges are used. The same is valid for discharge simulation when the estimated areal rainfall is implemented in hydrological modeling. However, the improvement is observed to be strongly dependent on the catchment characteristics, RC network density and spatial rain variability.

2) Because of the positive bias of the error distribution when using a log-transformed W-R relationship, areal rainfall overestimation is, in general, observed which resulted in an overestimation of discharges as well. This can be compensated by either increasing the RC network density or implementing more accurate optical sensors.

3) By increasing the rainfall measurement uncertainty by RCs, i.e. assuming larger variances for the random error, rainfall amount overestimation increases significantly. Implementing errors up to a certain level is observed beneficiary whereas larger uncertainties resulted in deterioration of results. Although the RCs with large errors should not be considered directly for rainfall measurement, relatively good *NSE* and *RMSE* values show the potential of RCs to be regarded as additional information in interpolation techniques.

4) It is observed that applying OK for areal rainfall estimation results in underestimation of rainfall amount. This was seen when no uncertainty was considered for RCs as well as for the case when only rain gauges were involved. Missing the rainfall maxima over the study area explains this phenomenon.

The hydrological simulations are carried out on hourly temporal resolution data with a lumped model parameter approach where the areal rainfall for each subcatchment is estimated separately. The conclusion of this study may not be valid for other cases when, for example, a distributed model or a different temporal resolution is being investigated. Depending on the target of each study, higher levels of data quality may be required. For instance, following the conclusions by Schilling (1991), in which he discussed implementing high spatial (1 km$^2$) and temporal (1 min) resolution data for urban hydrology, the quality of rainfall measurement by RCs might be insufficient.





Environmental factors such as road spray, car speed, wind direction and etc. can influence the performance of RCs in practice. Although this study showed that the RCs are beneficiary, field experiments are necessary to better assess the measurement uncertainty.

**Acknowledgment**

5   The study was funded by the German Research Foundation (DFG, 3504/5-2). The authors wish to thank Anne Fangmann and Sarah Louise Collins for their comments on an earlier draft of the paper.



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



**Table 1**. Number of cars driving at the same time for each RCs scenario on each catchment

|        | 5%  | 4%  | 3%  | 2%  | 1%  |
|--------|-----|-----|-----|-----|-----|
| Böhme  | 38  | 30  | 23  | 15  | 8   |
| Nette  | 138 | 110 | 83  | 55  | 28  |
| Sieber | 14  | 11  | 8   | 6   | 3   |





**Table 2**. Theoretical variogram model parameters used in this study, $a_{eff}$ is the effective range, $c_c$ the sill and $c_0$ the nugget effect

| Time period: | 2006 (01-03) | 2006 (04-09) | 2006-07 (10-03) | 2007 (04-09) | 2007-08 (10-03) | 2008 (04-09) | 2008-09 (10-03) | 2009 (04-09) | 2009-10 (10-03) | 2010 (04-09) | 2010 (10-12) |
|---|---|---|---|---|---|---|---|---|---|---|---|
| $c_0$ | 0.2 | 0.1 | 0.17 | 0.1 | 0.13 | 0.3 | 0.1 | 0.1 | 0.1 | 0.1 | 0.1 |
| $c_c$ | 1 | 0.9 | 0.9 | 0.85 | 1 | 0.7 | 1 | 0.85 | 1 | 0.87 | 1 |
| $a_{eff}$ | 60000 | 24000 | 42000 | 24000 | 42000 | 36000 | 48000 | 22500 | 45000 | 27000 | 48000 |


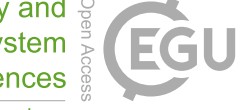

**Table 3**. Simulated discharge by rain gauges compared with the reference data

|        | Böhme | Nette | Sieber |
|--------|-------|-------|--------|
| RMSE   | 0.98  | 2.8   | 0.51   |
| NSE    | 0.95  | 0.76  | 0.86   |
| Pbias  | -6.2  | -22.5 | -15.8  |

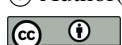



**Table 4.** Simulated discharge by RCs compared with the reference data

| Catchment | | | St. | 1% RCs | 2% RCs | 3% RCs | 4% RCs | 5% RCs |
|---|---|---|---|---|---|---|---|---|
| Böhme | | RMSE | 0.98 | 0.66 | 0.6 | 0.61 | 0.6 | 0.57 |
| | | NSE | 0.95 | 0.98 | 0.98 | 0.98 | 0.98 | 0.98 |
| | | Pbias | -6.2 | 6.4 | 6.1 | 7.4 | 7.6 | 7.4 |
| Nette | | RMSE | 2.8 | 1.01 | 0.84 | 0.67 | 0.73 | 0.76 |
| | | NSE | 0.76 | 0.97 | 0.98 | 0.99 | 0.98 | 0.98 |
| | | Pbias | -22.5 | 4.8 | 6.8 | 5 | 6.7 | 7.2 |
| Sieber | | RMSE | 0.51 | 0.55 | 0.58 | 0.53 | 0.48 | 0.37 |
| | | NSE | 0.86 | 0.83 | 0.81 | 0.84 | 0.88 | 0.92 |
| | | Pbias | -15.8 | 7.5 | 6 | 4.7 | 3.2 | 2.6 |





**Table 5**. Uncertainties for RCs when estimating areal rainfall and averaging over all subcatchments; 5% traffic model is considered

| | | $\sigma^2 = 0.0$ | $\sigma^2 = 0.01$ | $\sigma^2 = 0.04$ | $\sigma^2 = 0.09$ | $\sigma^2 = 0.021$ | St. |
|---|---|---|---|---|---|---|---|
| | RMSE | 0.27 | 0.27 | 0.30 | 0.38 | 0.28 | 0.44 |
| Böhme | NSE | 0.85 | 0.85 | 0.81 | 0.70 | 0.84 | 0.58 |
| | Pbias | -0.49 | 2.19 | 10.56 | 26.26 | 5.19 | -3.22 |
| | RMSE | 0.28 | 0.29 | 0.32 | 0.42 | 0.30 | 0.53 |
| Nette | NSE | 0.84 | 0.83 | 0.80 | 0.65 | 0.82 | 0.43 |
| | Pbias | -2.8 | 0.09 | 8.5 | 24.07 | 3.11 | -10.21 |
| | RMSE | 0.37 | 0.37 | 0.38 | 0.43 | 0.37 | 0.46 |
| Sieber | NSE | 0.69 | 0.69 | 0.68 | 0.6 | 0.69 | 0.53 |
| | Pbias | -4.55 | -1.7 | 6.4 | 21.5 | 1.2 | -14.45 |





**Table 6**. Investigating different uncertainties for RCs when simulating discharge compared with the reference discharge; 5% traffic model is considered

|  |  | $\sigma^2 = 0.0$ | $\sigma^2 = 0.01$ | $\sigma^2 = 0.04$ | $\sigma^2 = 0.09$ | $\sigma^2 = 0.021$ | St. |
|---|---|---|---|---|---|---|---|
| Böhme | RMSE | 0.34 | 0.38 | 1.05 | 2.64 | 0.57 | 0.98 |
|  | NSE | 0.99 | 0.99 | 0.94 | 0.65 | 0.98 | 0.95 |
|  | Pbias | -1 | 2.9 | 15.5 | 40 | 7.4 | -6.2 |
| Nette | RMSE | 0.65 | 0.52 | 1.56 | 4.19 | 0.76 | 2.8 |
|  | NSE | 0.99 | 0.99 | 0.92 | 0.45 | 0.98 | 0.76 |
|  | Pbias | -4.9 | 0.9 | 18.6 | 52.6 | 7.2 | -22.5 |
| Sieber | RMSE | 0.37 | 0.36 | 0.43 | 0.71 | 0.37 | 0.51 |
|  | NSE | 0.93 | 0.93 | 0.9 | 0.72 | 0.92 | 0.86 |
|  | Pbias | -4.5 | -1 | 9.1 | 28 | 2.6 | -15.8 |





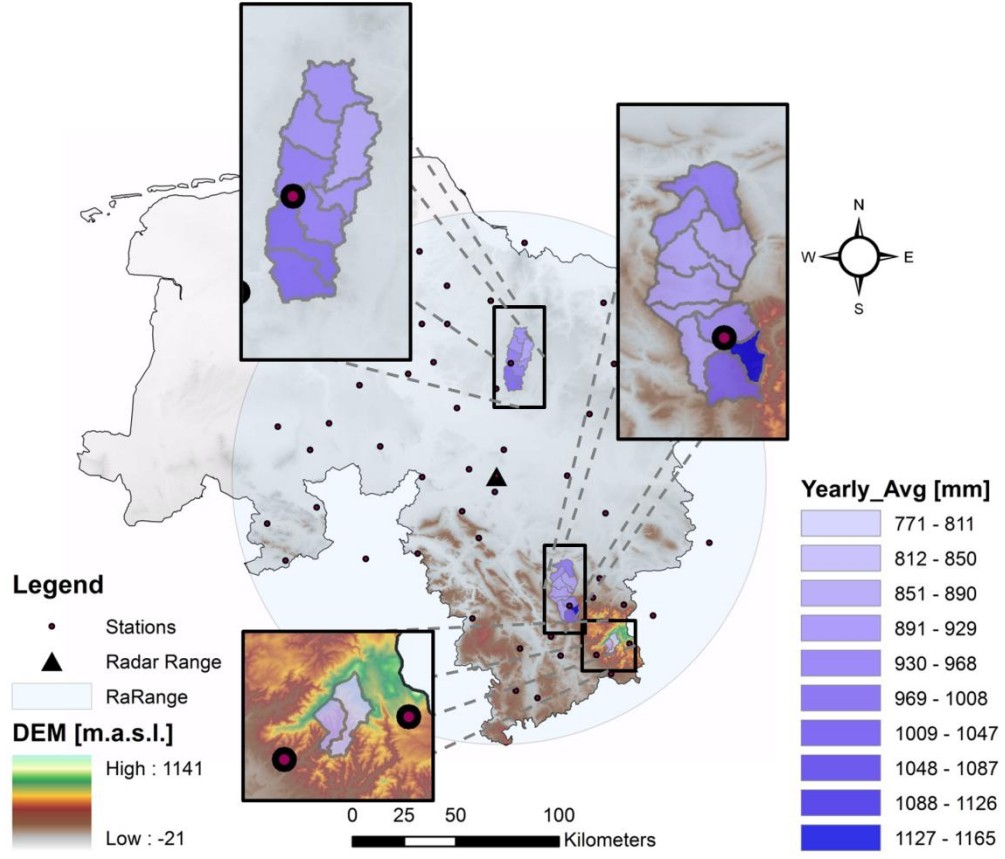

**Fig. 1.** Study area, catchments, station network and mean annual precipitation (MAP) from 2006 to 2010. From north to south: Böhme, Nette and Sieber catchments.

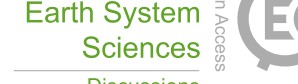

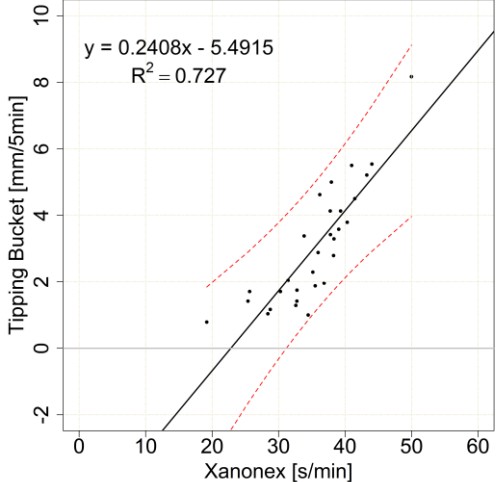

**Fig. 2.** Xanonex W-R relationship (Rabiei et al., 2013)





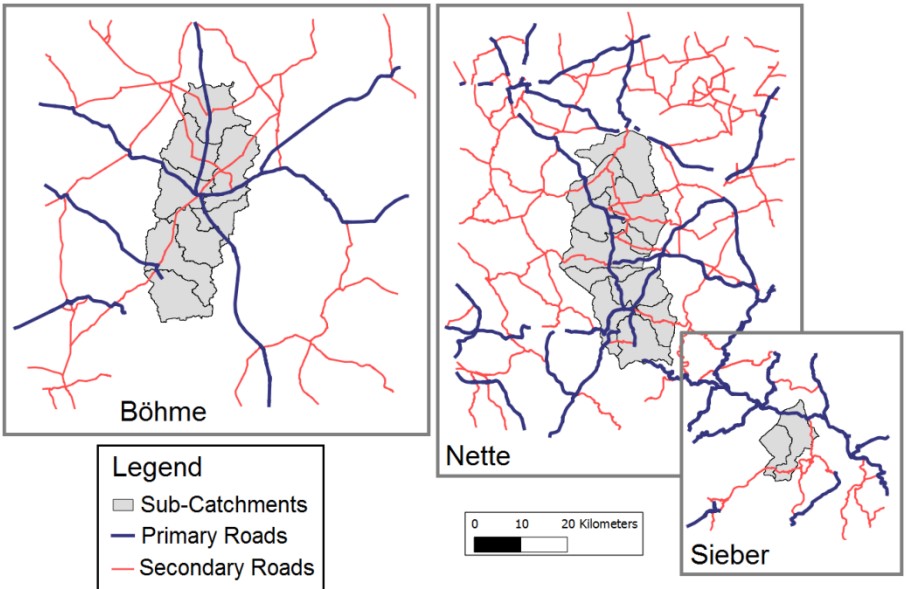

**Fig. 3**. The road network on which the RCs are modeled.





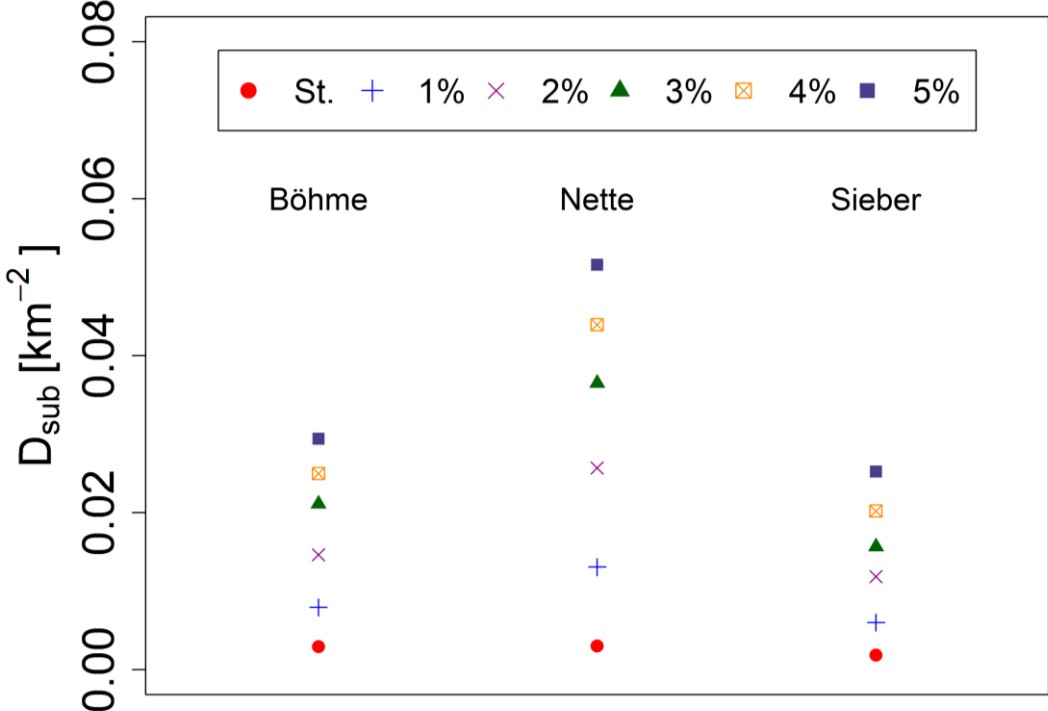

**Fig. 4.** Network density of the catchments for different scenarios





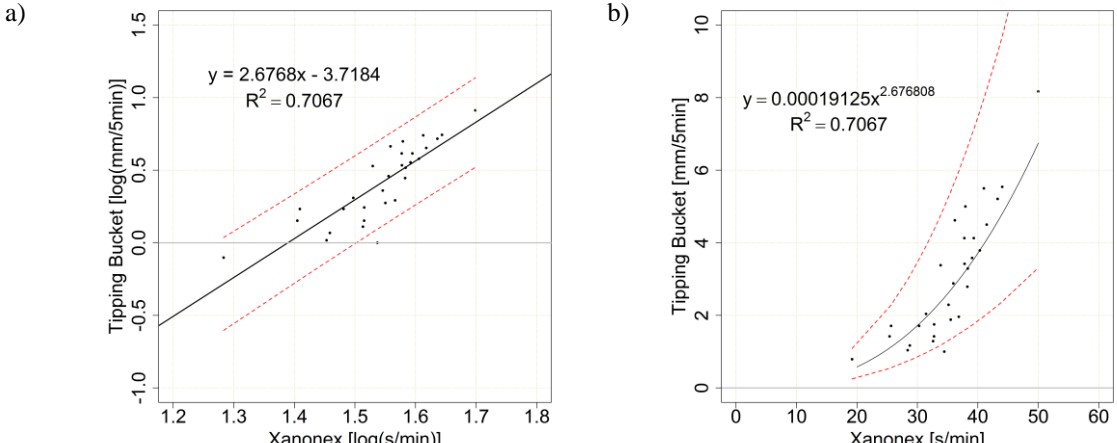

**Fig. 5.** (a) Xanonex log transformed W-R relationship (b) Xanonex W-R relationship after Eq. 5; the red lines illustrate the 95% prediction limits





**Fig. 6**. Areal rainfall estimation using rain gauges compared with the reference data in, from left to right, the Böhme, Nette and Sieber catchments. The polygons toward green colour present better results than the ones toward red colour.





**Fig. 7**. Areal rainfall estimation evaluation using RCs for the Böhme catchment. The green colour for *RMSE$_{diff}$* and *NSE$_{diff}$* illustrate the improvement of the areal rainfall estimation quality when RCs are used compared with when only rain gauges are considered.





**Fig. 8**. Areal rainfall estimation evaluation using RCs for the Nette and Sieber catchment. The green colour for *RMSE*$_{diff}$ and *NSE*$_{diff}$ illustrate the improvement of the areal rainfall estimation quality when RCs are used compared with when only rain gauges are considered.



**Fig. 9**. Discharge simulation using different sources of data for the Nette catchment. For RCs only 5% scenario is
considered.