# Peer review of "Areal rainfall estimation using moving cars – computer experiments including hydrological modeling"

_Hydrology and Earth System Sciences, 2016_

## Referee Comment (RC1) · G. Pegram (Referee) · 2 Mar 2016

I was drawn to this paper by what seemed to me to be an interesting topic, but was disappointed by what I found to be a rather naïve desktop experiment, using some unsupported assumptions and severe simplifications. The paper seems to be the second or third in a series on the topic by the first two authors and I'm not sure that there is much originality in this one. Sadly, I came away underwhelmed by the conclusions and the usefulness of the procedures suggested and am not convinced that the idea is do-able, nor practical. For one thing, I find it difficult to imagine that enough car/lorry owners would install and maintain the sensor and transponder in their vehicles

to make the density of the instruments useful. Do many vehicles have variable speed wipers? Does vehicle speed and shape not have an effect? In addition, seasonality and diurnal variations were not considered - what do you do with snow? - how do you measure at night when traffic density is low? - what about spray from cars ahead on wet roads when it is not raining? what about wind? how do you use the RC data when a car travelling at 120 k/h moves 10 km within 5 minutes between radar scans? These practical problems would seem to overwhelm the conclusions drawn from the laboratory/computer experiments.

My opinion is that the paper has been exposed in HESSD but do not think it is good enough to be published in HESS. My remarks on a few individual issues follow below.

Geoff Pegram 02 March 2016

5: 14 I argue with the unjustified assumption that "Due to the long period of time considered in this study, the day-night variation in traffic count is considered insignificant."

6: 27 as written, the error epsilon being multiplicative could cause negative and zero rainfall values; replace by (1 + epsilon) - also if left as is, equ (6) is likely to attempt to compute the logarithm of a negative number

7: 10 The assumptions of the relationships are not supported by 'data' nor curves. How does one know that the formulation in Eq (6) is valid? How do you differentiate between rain and snow?

7: 15 ff how skew are the data - OK assumes Gaussianity, so some transform might be required

8: 4 "An exponential variogram is considered as the theoretical variogram model:" there is no justification for this choice based on data

10:27 explain the combination of degree and kilometre as a radar-based spatial measure

15: 13 "As seen in the study area and data section," what does this mean?

15: 24 Why not include a discussion of Fig 8 in this section? It's the same subject as Fig 7 and the referencing of these figures is scattered in this and the next section

19:19 poorLY

19: 20 "By increasing the uncertainties, i.e. enlarging sigma-squared , the overestimation of rainfall amount affects the model performance as well." What do you mean by this last sentence? Surely one does not just 'enlarge' sigma? What is a realistic value? how is the model performance affected? what is being emphasised by the blue ovals magnified in Figure 9?

20: 10 quality of MODELLED areal

20: 15 can't you find a suitable transform?

20: 24 possibly try a Gaussian transform of the data ?

21: 1 remove 'and' and add 'snow, night/day variations . . .' ?

---

## Referee Comment (RC2) · Anonymous Referee #2 · 12 Mar 2016

**Review of the manuscript HESS-2016-17:**
**"areal rainfall estimation using moving cars - computer experiments including hydrological modeling"**
**by E. Rabiei, U. Haberlandt, M. Sester, D. Fitzner, M. Waliner**

March 12, 2016

**Summary**

This manuscript presents a simulation study to evaluate the potential benefit of using information derived from sensors on-board moving cars to improve the estimation of areal rain rate over small (sub-)catchments for hydrological modeling. Because of the difficult access and limited number of actual data from moving cars (denoted RC), a simulation approach is employed, assuming car density and movement along main and secondary roads within 3 catchments in southern Germany. The reference areal rainfall is derived from an operational weather radar, in order to have realistic spatial patterns. The contribution of RCs is analyzed by comparing the quality of the retrieved areal rainfall and simulated discharge over the three catchments with results obtained

using rain rate from rain gauges (from 1 in the vicinity to 2 within the catchment) only. The added-value of using RCs (at varying densities) is demonstrated in terms of areal rainfall and discharge simulation in situation where the rainfall may vary significantly within the catchment.

**Recommendation**

The topic is original and relevant to the hydrological community focusing on small scale catchments. The task is difficult because many influencing factors must be realistically simulated in order to obtain meaningful results. I think the authors do a good job to this respect, although some of the assumptions or simplifications should be more clearly mentioned and better explained. The question of the transferability of the obtained results and conclusions remains open and should be better tackled.

Overall, there are a few issues that need to be properly addressed in order to have a manuscript acceptable for publication in HESS. I provide a list of comments and questions below.

**General comments**

1. Rainfall is a complex process having many facets. What this work is about is rainfall intensity or rain rate estimation, but there are other aspects of rainfall that may be relevant to different communities (e.g. microphysics). I hence suggest that rain rate is used instead of rainfall throughout the manuscript when this is what is of interested. Starting in the title...

2. Concerning performance assessment for areal rain rate, an important aspect of rainfall that is not mentioned in the manuscript is intermittency. In particular when

working at a temporal resolution of 5 min, there will be a significant proportion of the catchment where there is no rain. The employed evaluation criteria do not take this into account. I suggest to evaluate the occurrence and the quantity separately.

3. Still concerning rainfall, using ordinary kriging requires some important assumptions, not mentioned: the data must be Gaussian for the optimality and rain rate is far from Gaussian; the random process of interest must be stationary and this is likely not the case in the mountainous catchment (the influence of topography is mentioned in section 4 but not the issue with ordinary kriging); depending on the employed variogram model, negative values can be generated, if it is the case the way they are treated should be explained.

4. I did not understand if the areal rain rate and the discharge values were continuously estimated over the 5 years or if the analysis was on an event basis. In particular in the results displayed in Figure 6, 7 and 8. A point to clarify as if a continuous approach is employed, there will be mostly no rain over the considered catchments (in the order of 10-15% of the time steps are rainy I imagine).

5. The results presented and analyzed in this work are specific to the considered catchments. Therefore, the question of the transferability or generalization of the conclusions can be raised. I recommend the authors to clarify which conclusions may hold for other regions and which ones may not, with appropriate explanations of course.

**Specific comments**

1. P.2, l.15: maybe the spaceborne radar of the GPM mission could be mentioned, as there are active remote sensing instruments in orbit.

2. P.2 and 3: the scientific questions 1 and 2 read quite similar to me, the difference should be clarified in my opinion.

3. P.4, section 2.1: it was not clear to me while reading this section if the rain gauge data were also extracted from the radar image. This is clarified in the conclusions, but I think it should be done earlier.

4. P.5, item b: rain rate will influence the speed of cars via visibility. It becomes very difficult to drive when the rain rate is above 40-50 mm/h...

5. P.5, l.14-15: I do not understand why the day-night cycle does not influence the estimation of rain rate by RCs, as much less cars are to be expected during night. This point should be clarified.

6. P.5, Eq.3: what is the value used for $r$?

7. P.6, l. 15-19: what do these sensors exactly measure? What is the signal coming out of those? Not essential, but out of curiosity...

8. P.7, l.5: it should be mentioned that the parameters of the linear regression on the log are optimal in the log space but not in the (original) linear space, giving more weight to small R values.

9. The normalization proposed in Eq.8 can be called "climatological variogram" (*Lebel and Bastin*, 1985).

10. P.9, Eq.11-13: RMSE is a dimensional quantity, so its units should be provided. More generally, the units (also for dimensional quantity like Pbias - in %?)  are missing in many part of the text, this should be corrected.

11. P.10, l.10: from which data is the mean annual rainfall derived?

12. P.11, l.15-16: the values of $a$ and $b$ should also be given here, not only in Fig. 5.

13. P.13, l.10: the goodness of fit of the variogram model should be provided to give the reader a sense of the accuracy of the fit.

14. P.14, l.23-24: as mentioned in the general comments, this sentence clearly refers to non-stationarities, which hence question the use/applicability of ordinary kriging.

15. P.15, l.17: it is a matter of detail, but it cannot be concluded from the data at hand that the discharge will be dominated by these 2 sub-catchments, but only assumed. Other factors than rain rate may influence discharge amount (geomorphology, karst,...).

16. P.15, l.24: RMSE is an absolute value so it cannot be compared between the three catchments.

17. P.18, l.1-2: I appreciate the tentative to generalize the results, but these statements are very speculative. This comment is in the line with item 5 in the general comments.

18. P.18, l.21-22: I do not understand this sentence.

19. P. 19, l.1: this refers to the positive skewness of the distribution of rain rate, that could be mentioned here.

20. P.20, l.29: a reference to the more recent paper by *Berne et al.* (2004) on the same topic could be added here.

21. Tables 2-6 and Figures 6-8: units should be provided.

22. Fig.1: the lowest altitude in the color bar is negative, it should be to 0 (or more) I guess for this region of Germany.

23. Fig.5: I am surprised by the identical correlation coefficient in both panels...

24. Figures 7-8: when dealing with differences for which 0 matters, I suggest to use another color bar (blue for negative, red for positive, white for 0) which would ease the identification of the global pattern of under- or over-estimation.

**References**

Berne, A., G. Delrieu, J.-D. Creutin, and C. Obled (2004), Temporal and spatial resolution of rainfall measurements required for urban hydrology, *J. Hydrol.*, *299*(3-4), 166–179.

Lebel, T., and G. Bastin (1985), Variogram identification by the mean-square interpolation error method with application to hydrologic fields, *J. Hydrol.*, *77*(1-4), 31–56.

---

## Author Comment (AC1) · 12 Apr 2016

**Areal rainfall estimation using moving cars – computer experiments including hydrological modeling**

Ehsan Rabiei[1], Uwe Haberlandt[1], Monika Sester[2], Daniel Fitzner[2], Markus Wallner[3]

[1] [Institute of Water Resources Management, Hydrology and Agricultural Hydraulic Engineering, Leibniz Universität Hannover, Hanover, 30167, Germany]

[2] [Institute of Cartography and Geoinformatics, Leibniz Universität Hannover, Hanover, Germany]

[3] [Federal Institute for Geosciences and Natural Resources, Groundwater Resources - Quality and Dynamics, Hanover, 30655, Germany]

*Correspondence to*: E. Rabiei (rabiei@iww.uni-hannover.de)

**GENERAL REMARKS (BOTH REFEREES):**

We are very grateful to G. Pegram and an anonymous referee for their remarks on our manuscript. We believe that considering all the points will improve the quality of the manuscript significantly.

In the following we will respond all the comments, separately. All the numbers corresponding to figures and tables in this document refer to the numbers used in the original manuscript provided for HESSD.

**GENERAL REMARKS OF REFEREE #1:**

We agree about the problems related to RainCars, in particular for practical purposes. Some of them are addressed in the text and many of them are discussed in previous publications (Fitzner et al., 2013; Rabiei et al., 2013). We have now included a brief discussion about the problems in the text. The main objectives of this study are using errors derived from laboratory experiments (Rabiei et al., 2013) and investigating scenarios with larger uncertainties considering the problems in field experiments, to assess the value of RainCars. The message of the paper is that the RCs would be beneficial when errors lay below a certain threshold and the car density is above a certain level. The practicability of the idea can only be assessed after carrying out extended field experiments which would be a necessary future step. Then, the errors from field experiments could be compared against the thresholds obtained here to decide about feasibility of the approach.

Finding volunteers for field experiment is hard. That is the reason we cooperated with taxi companies in Hanover in previous years. Considering the fact that more than 53000 taxis were available in Germany in 2012 (http://de.statista.com/statistik/daten/studie/36475/umfrage/anzahl-der-taxen-in-deutschland-seit-1960/), one realizes the huge potential of using public transportation as moving sensors. Using wiper frequency for rainfall measurement was the initial idea which we changed later to using the move objective optical sensors for rainfall measurement. The diurnal variation of the

number of RCs was considered and discussed by Haberlandt and Sester (2010). Here, instead, the RC density scenarios from 1% to 5% could be used to assess different traffic densities during day and night. A possible method using RCs for practical purposes was proposed by Fitzner et al. (2013). In that study, a communication between the moving sensors and rain gauge network was proposed. This could in practice filter some of the errors out. Additionally, large rainfall amounts caused by front spray may be detected using statistical approaches. However, we agree that front spay is one of the main problems which must be considered when more data from field experiment is available. Wind is an important factor influencing under- or overestimation of rainfall by RCs and could be compensated in a similar way to car speed. A theoretical relationship for compensating car speed was suggested by Rabiei et al. (2013) which was tested in practice by Fitzner et al. (2013). Those findings could be used to compensate the influence of both wind and car speed.

There is no need to connect RC data with radar data. These would be two independent sources of data. However, the merging product of all available data sources might provide a more realistic rainfall field map. Considering the facts that we are able to record all the sensor readings every second and the assumed *conservative* car speed of 80 km/h, RCs would move up to 25 meters every second which give useful rainfall observations. Cars on highways are not used here because of the high speed and heavy road spray.

**SPECIFIC COMMENTS**

**5: 14** We agree that this is an important factor. As explained, different RC density scenarios could address the diurnal fluctuation. This is added to the manuscript.

**6: 27** (log ε) is assumed to be normally distributed and can be negative. After back transformation, $\varepsilon = 10^{\log\varepsilon}$, the error (ε) cannot be negative in Eq. (6).

**7: 10** A linear relationship was proposed by Rabiei et al. (2013) in order to find a relationship between sensor readings and rain rate. Due to all the difficulties associated to this relationship, we decided to implement a simple data transformation to overcome the problems. One major concern was when the surface of the optical sensor is dry or at the vicinity of the origin. Not only the curve must pass through the origin, but also the rain rate at this point using the W-R relationship (1) must not be negative and (2) must be the most accurate value. Therefore, by taking the logarithm from both sides these two concerns would be satisfied. We agree that other transformations might be more suitable, but the simplicity of the method was also an important point considered in this step.

**7: 15** We agree that implementing OK, in particular for fine temporal resolution, is questionable, but this approach is carried out in several studies. Although applying data transformation might overcome this theoretical issue, the results might still be very similar to the current results due to the fact that the main problem is rather the distance between the observations (network density). However, as comparing different sources of data is carried out using the same technique, this issue is relatively insignificant in this study. A summary of what is explained here is added to the text.

**8: 4** It was decided to provide the variogram properties in a table instead. Following figures illustrate the variograms used in this study. The numbers illustrate different time windows considered due to seasonal change in variogram properties. Due to the relative comparison of the scenarios, the figures are not provided in the text.

**2006 (01 - 03)**

[Figure]

**2006 (04 - 09)**

[Figure]

**2006 (10) - 2007(03)**

[Figure]

**2007 (04 - 09)**

[Figure]

**2007 (10) - 2008(03)**

[Figure]

**2008 (04 - 09)**

[Figure]

**2008 (10) - 2009(03)**

[Figure]

**2009 (04 - 09)**

[Figure]

[Figure]

[Figure]

[Figure]

**10: 27** We agree that the description is not comprehensive. A better description of weather radar data is provided in the text.

**15: 13** Here, the main purpose was to draw the reader's attention to the spatial rainfall variation explained earlier. To be clearer, the text is changed.

**15: 24** It was decided to explain different sources of data, separately. Fig. 7 and Fig. 8 are provided in the same subsection which is discussed on page 16.

**19: 19** Yes, the word is changed accordingly.

**19: 20** Sigma was considered as a measure illustrating the magnitude of the device uncertainty. In addition to addressing the uncertainty derived from the laboratory experiment, we decided to investigate what happens if we face higher uncertainties. The large uncertainty could come from all the difficulties mentioned earlier such as wind. See also response to general remarks

The oval was used to make it easier for the reader to see the differences between the scenarios. A new figure is replaced illustrating the discharge over a shorter time period.

**20: 10** Yes, MODELLED is added to the text

**20: 15** please refer to the answer to comment 7:10. This transformation was implemented given all the points mentioned above. Another reason of implementing this transformation was the ease in finding the range for error, Fig. 6a.

**20: 24** We agree that transforming the data in a way that they fit the Gaussian distribution function follows the assumptions behind OK, but implementing OK on rainfall observation data is a popular technique used for areal rainfall estimation in several studies. This method was considered for all sources of data, so a relative comparison is feasible.

**21: 1** The text is changed accordingly.

**GENERAL REMARKS OF REFEREE #2:**

1. We agree that the term rain rate is a better term especially as we worked with radar data. Due to the fact that the data is used in a hydrological model, we decided to used rainfall estimation which is a term containing both rain rate and rainfall amount. Depending on the intention of each sentence, a more appropriate term is replaced. The term rainfall amount is omitted from the text.

2. Yes, we agree. However, as the hydrological model used in this study has a semi-distributed approach and the subcatchments are relatively small, the separated evaluation of the subcatchments may represent the sought information.

3. Yes, we agree that OK is only optimal when the data are Gaussian. Nevertheless, implementing OK is used in several studies without applying any transformation. However, the results may not be changed significantly even if transforming the data would satisfy the theoretical assumptions. The main problem, as it is discussed, is the number of observations. In other words, more information is required for a better areal rainfall estimation rather than Gaussianity of data under study. The part discussing this problem is added to the text. See also the answer to the comment of referee #1 (20: 24).

4. This was pointed out on p.2 L31-L32: "*A continuous investigation using RCs with the derived uncertainties from laboratory experiments for a long period of time as well as implementing the data in a hydrological model would answer three important scientific questions*". As explained in the text, a continuous approach was used. Yes, like any continuous approach there are time steps with no rain.

5. Yes, we agree that the conclusion of this study may not be valid overall. However, it is a new approach to investigate whether additional information of RCs would be useful. A better discussion is added to the text.

**SPECIFIC COMMENTS**

1. The terms describing the method is added to the text.
2. The second scientific question is explaining the reason of investigating several other uncertainties for RCs. This part is rephrased.
3. The text is changed accordingly.
4. Yes, car speed also causes overestimation of rain rate. As explained earlier, a theoretical relationship for compensating car speed was suggested by Rabiei et al. (2013) which was proved in practice by Fitzner et al. (2013). Those findings could be used to compensate the influence of both wind and car speed. A part is added to explain this approach for compensating car speed.
5. Please refer to the answer to the first comment of the first referee (SPECIFIC COMMENTS).
6. A constant value of 20000 m is used. This value is added to the text.
7. This was explained in details in Rabiei et al. (2013). A short summary is added to the text.
8. This explanation is added to the text.
9. The term is added to the text.
10. All the units are added to the text.
11. This is from ground observation and was mentioned in Berndt et al. (2014)
12. Those values are added to the text.
13. Please refer to the answer to comment 8) of the first referee.
14. Yes, we agree and it was discussed earlier here.
15. Yes, we agree. The term "can" is replaced by "may". A better discussion is added to this part.

16. Yes, RMSE is deleted from the sentence.
17. The sentences are rephrased to describe the situation only for this study.
18. The first sentence refers to the possible ways of using RCs even if the uncertainty is relatively large. The beginning of the new paragraph represents the situation when the RCs are as accurate as ground observations. The underestimation of rain rates were observed even in this situation.
19. Yes, this is added to the text.
20. The new citation is added to the text.
21. The units are added to the tables and figures.
22. Yes, this was generated automatically. The lower altitude is set now to zero.
23. The two figures represent the same relationship. On the left it is provided in a logarithmic scale and on the right after transforming back. This is the reason of having the same $R^2$ value. On the right a power relationship describes the relationship between the two variables, whereas the left figure is a linear relationship in logarithmic scale.
24. The colors are changed for the ease of identifying the global pattern.

**References**

Berndt, C., Rabiei, E., Haberlandt, U., 2014. Geostatistical merging of rain gauge and radar data for high temporal resolutions and various station density scenarios. Journal of Hydrology, 508(0): 88-101.

Fitzner, D., Sester, M., Haberlandt, U., Rabiei, E., 2013. Rainfall Estimation with a Geosensor Network of Cars Theoretical Considerations and First Results. Photogrammetrie - Fernerkundung - Geoinformation, 2013(2): 93-103.

Haberlandt, U., Sester, M., 2010. Areal rainfall estimation using moving cars as rain gauges – a modelling study. Hydrol. Earth Syst. Sci., 14(7): 1139-1151.

Rabiei, E., Haberlandt, U., Sester, M., Fitzner, D., 2013. Rainfall estimation using moving cars as rain gauges - laboratory experiments. Hydrol. Earth Syst. Sci., 17(11): 4701-4712.

---

## Referee Report (RR1)

**Review of the revised version of the manuscript HESS-2016-17: "Areal rainfall estimation using moving cars - computer experiments including hydrological modeling" by E. Rabiei, U. Haberlandt, M. Sester, D. Fitzner, M. Waliner**

**Recommendation**

I have to say that I am a bit disappointed by the revised version of this manuscript, in the sense that the authors did not properly address some of the important questions and issues raised by the two reviewers. I further elaborate on this issue in thenext section.

I do not think the manuscript is suitable for publication in HESS as is, and in my view major revisions are still needed to reach the required quality.

**General comments**

1. I will not argue on vocabulary, but in some part of the text, the use of "rainfall estimation" is confusing in my opinion. Not a major issue though...

2. I understand that the authors do not want to address the issue of rainfall intermittency, but at least this should be mentioned in the text. Again, this could be a significant source of uncertainty in areal rain-rate estimation when working at relatively high temporal resolution like 5 min. In particular, the authors do not explain how they treat 0 in rain rate values.

3. Same here: it is not sufficient to say that other studies do the same mistake by applying kriging to non-Gaussian data to wave out the issue. In addition, the authors did not address the other part of my comment concerning stationarity and the treatment of possible negative rain-rate values. Stationarity is a crucial assumption for kriging and it is not even mentioned in the paper... By not separating the occurrence and the quantity of rainfall in the interpolation, there is a risk to generate negative values of rain rate. How this issue is addresed is not mentioned either.

4. It seems the authors missed the point of my questions about continuous simulation. If I am getting it correct, they continuously simulate 4-5 years of rain rate and the corresponding discharge over different catchments. If so, the time series of areal rain rate and discharge values will be dominated by 0 and low values respectively. Such dominance may affect the estimated comparison criteria and bias the intepretation conducted on these results (i.e. fig.6 and 7). This was my question in the first round of review...

**Specific comments**

1. P.5, l.24: "proposed to use..." instead of "proposed using...".

2. P.6, l.24-25: the added sentence explaining how the sensor is working reads strange (e.g., bounce, escape...).

3. P.7, l.7-8: "Considering Eq.4 does not...": the sentence is incomplete or incorrect.

4. P.7, l.17: obtained instead of estimated.

5. P.13, l.15: preventing the derivation (just a suggestion) but estimation is not relevant here.

6. P.20, l.4: "and that we did not take into...".

7. P.20, l.23: rainfall spatial variability.

8. P.21, l.3: the underestimation likely results from the non-Gaussianity of the rain rate, as kriging is an unbiased estimator when the required assumptions are met.

---

## Referee Report (RR2)

[referee-annotated manuscript omitted]

---

## Referee Report (RR3)

I am satisfied with the changes implemented by the authors related to my previous comments, except number 4.

This is not crucial, but it seems the authors do not get my point. I am not criticizing the fact of performing continuous simulations in time, I just mention that because it does not rain most of the time, there will be a lot of 0 rain-rate values that will "bias" the criteria used (RMSE, NSE,...). It is not critical in the sense that these criteria are used in a relative sense, but their absolute values (of the criteria) are likely not representative of the performance during rainfall events (which is the goal of the proposed approach using RCs).

Just two very minor suggestions:
1. P.14, l.6: I guess the bracket should be after "month" and before "intervals".
2. P.26, caption Table2: I suggest to add "exponential" between "Theoretical" and "variogram".

---

## Author Response (AR2)

**Areal rainfall estimation using moving cars – computer experiments including hydrological modeling**

Ehsan Rabiei[1], Uwe Haberlandt[1], Monika Sester[2], Daniel Fitzner[2], Markus Wallner[3]

[1] [Institute of Water Resources Management, Hydrology and Agricultural Hydraulic Engineering, Leibniz Universität Hannover, Hanover, 30167, Germany]

[2] [Institute of Cartography and Geoinformatics, Leibniz Universität Hannover, Hanover, Germany]

[3] [Federal Institute for Geosciences and Natural Resources, Groundwater Resources - Quality and Dynamics, Hanover, 30655, Germany]

*Correspondence to*: E. Rabiei (rabiei@iww.uni-hannover.de)

**GENERAL REMARKS (BOTH REFEREES):**

We are very thankful to G. Pegram and an anonymous referee for their remarks on the revised manuscript. We believe that considering all the remaining unclear points will improve the quality of the manuscript.

In the following, we will respond all the comments, separately. All the numbers corresponding to figures and tables in this document refer to the numbers used in the manuscript provided for HESSD after the first revision.

**GENERAL REMARKS OF REFEREE #1:**

We thank the first referee for all the detailed remarks provided in the text. All the remarks are either modified following the provided suggestions, or a better explanation is added for unclear parts.

**GENERAL REMARKS OF REFEREE #2:**

1.  We agree that the use of "rainfall estimation" should be changed in some parts. The phrase is changed to "rain intensity" if required.
2.  Yes, we agree that the spatial and tempral variability of rainfall becomes significant especially in fine temporal resolutions. The whole idea behind RainCars is to increase the number of observations in order to overcome this issue. This is added to the text to make the issue clear. 0 values were not changed and according to the properties of the used variograms, they influenced a certain area in its vicinity like other values.
3.  Yes, we agree that non-stationarity of the data may cause problems. In order to compensate this problem, only a certain number of stations is used for rain-rate estimations. This means that only the some of the neighboring extracted values are (for RCs or rain gauges) used for

rain-rate estimation. This may explain the reason behind implementing OK considering the non-stationarity of data. Moreover, as is provided in Table 2, the nugget effect exists in all the used variograms. This decreases the possibility of facing negative weights in Kriging matrix, and consequently, negative rain-rate estimations. However, in rare occasions when facing negative estimations, the values are set to zero. This is added to the text.

4. Yes, we have some time steps with no rain. However, this is what one faces in practice. We decided to evaluate the RainCars in an impartial comparison.

**ESPECIFIC COMMENTS OF REFEREE #2:**

1. The phrase is changed accordingly.
2. The added sentence is modified.
3. This sentence is rephrased.
4. The word "obtained" is replaced.
5. The word "derivation" is replaced.
6. The sentense is rephrased.
7. The phrase is changed.
8. This is added to the text.

[revised manuscript text omitted]